

**Leaf phenology as one important driver of seasonal changes in isoprene emission in**
**central Amazonia**
*Eliane G. Alves[1], Julio Tóta[2], Andrew Turnipseed[3], Alex B. Guenther[4], José Oscar W.*
*Vega Bustillos[5], Raoni A. Santana[2], Glauber G. Cirino[6], Julia V. Tavares[1], Aline Lopes[1],*
*Bruce W. Nelson[1], Rodrigo A. de Souza[7], Dasa Gu[4], Trissevgeni Stavrakou[8], David K.*
*Adams[9], Jin Wu[10], Scott Saleska[11], Antonio O. Manzi[12].*
[1] Department of Environmental Dynamics, National Institute for Amazonian Research
(INPA), Av. André Araújo 2936, CEP 69067-375, Manaus-AM, Brazil.
[2] Institute of Engineering and Geoscience, Federal University of West Para (UFOPA),
Rua Vera Paz s/n, CEP 68035-110, Santarem-PA, Brazil.
[3] 2B Technologies, Inc., 2100 Central Ave., Boulder, CO 80301, U.S.A.
[4] Department of Earth System Science, University of California, Irvine, CA 92697, USA.
[5] Chemistry and Environment Center, National Institute for Energy?? and Nuclear
Research (IPEN), Av. Lineu Prestes 2242, CEP 05508-000, São Paulo-SP, Brazil.
[6] Department of Meteorology, Geosciences Institute, Federal University of Para, Belém,
PA 66075-110, Brazil
[7] Meteorology Department, State University of Amazonas (UEA), Av. Darcy Vargas
1200, CEP 69050-020, Manaus-AM, Brazil.
[8] Royal Belgian Institute for Space Aeronomy, Avenue Circulaire 3, 1180, Brussels,
Belgium.
[9] Centro de Ciencias de la Atmósfera, Universidad Nacional Autónoma de México, Av.
Universidad 3000, 04510, Mexico city, Federal District, Mexico.



[10] Department of Environmental and Climate Sciences, Brookhaven National Laboratory,
Upton, NY 11973, USA.
[11] Ecology and Evolutionary Biology Department, University of Arizona, Cherry Avenue
and University Boulevard, Tucson, AZ 85721, USA.
[12] National Institute for Spatial Research, Center of Weather Forecasting and Climate
Studies, Rod. Presidente Dutra, km 40, Cachoeira Paulista-SP, Brazil.
**corresponding author:** +55 92 3643 3238; elianegomes.alves@gmail.com
**Abstract**
Isoprene fluxes vary seasonally with changes in environmental factors (e.g., solar
radiation and temperature) and biological factors (e.g., leaf phenology). However, our
understanding of seasonal patterns of isoprene fluxes and associated mechanistic controls
are still limited, especially in Amazonian evergreen forests. In this paper, we aim to
connect intensive, field-based measurements of canopy isoprene flux over a central
Amazonian evergreen forest with meteorological observations and with tower-camera
leaf phenology to improve understanding of patterns and causes of isoprene flux
seasonality. Our results demonstrate that the highest isoprene emissions are observed
during the dry and dry-to-wet transition seasons, whereas the lowest emissions were
found during the wet-to-dry transition season. Our results also indicate that light and
temperature can not totally explain the isoprene flux seasonality. Instead, the camera-
derived leaf area index (LAI) of recently mature leaf-age class (e.g. leaf ages of 3-5
months) exhibits the highest correlation with observed isoprene flux seasonality
($R^2$=0.59, $p$<0.05). Attempting to better represent leaf phenology in the Model of





Emissions of Gases and Aerosols from Nature (MEGAN 2.1), we improved the leaf age
algorithm utilizing results from the camera-derived leaf phenology that provided LAI
categorized in three different leaf ages. The model results show that the observations of
age-dependent isoprene emission capacity, in conjunction with camera-derived leaf age
demography, significantly improved simulations in terms of seasonal variations of
isoprene fluxes ($R^2$=0.52, $p$<0.05). This study highlights the importance of accounting for
differences in isoprene emission capacity across canopy leaf age classes and of
identifying forest adaptive mechanisms that underlie seasonal variation of isoprene
emissions in Amazonia.

**1. Introduction**

Isoprene is considered the dominant contribution to Biogenic Volatile Organic

Compound (BVOC) emission from many landscapes and represents the largest input to
total global BVOC emission, which has the magnitude of 400-600 Tg C y$^{-1}$ ( see Table 1
of Arneth et al., 2008). This compound regulates large-scale biogeochemical cycles. For
example, once in the atmosphere, isoprene has implications for chemical and physical
processes due to its reactivity, influences on the atmospheric oxidative capacity, as well
as its potential to form secondary organic aerosols (Claeys et al., 2004), which interact
with solar radiation and act as effective cloud condensation nuclei. Moreover, carbon
dioxide is believed to be the fate of almost half of the carbon released in the form of
BVOCs (Goldstein and Galbally, 2007) and, as BVOC emissions are regarded as highly
significant for ecosystem productivity (Kesselmeier et al., 2002) with isoprene being the
most emitted hydrocarbon, it thereby plays an important role in carbon balance.



Tropical forests are the largest source of isoprene to the atmosphere, contributing
almost half of the estimated global annual isoprene emission, according to Model of
Emissions of Gases and Aerosols from Nature (MEGAN) estimates (Guenther et al.,
2006). Given that the Amazon basin is the largest territorial contribution to global
tropical forests, this ecosystem is thought to be one of the most important sources of
isoprene to the global atmosphere.
Recently, remotely sensed observations of multiple years have revealed seasonal
changes in isoprene emission over the Amazonian rainforest (Barkley et al., 2008, 2009,
2013, Bauwens et al., 2016). Apart from these remotely sensed data, only a few studies
based on *in situ* data exist (Alves et al., 2016; Andreae et al., 2002; Kesselmeier et al.,
2002; Kuhn et al., 2004b; Yáñez-Serrano et al., 2015). Some of these *in situ* studies
indicate that environmental factors such as solar radiation and temperature are primary
drivers of isoprene (Andreae et al., 2002; Kesselmeier et al., 2002; Kuhn et al., 2004b;
Yáñez-Serrano et al., 2015).
Canopy phenology has been suggested as the primary cause of seasonal changes
of photosynthesis in Amazonian ecosystems (Wu et al., 2016), a suggestion in agreement
with reports on phenology effects at the Amazonian tree species level (Kuhn et al.,
2004a). Given that photosynthesis provides substrates and energy for isoprene production
and that isoprene is not stored within leaves, canopy phenology could therefore be an
important seasonal driver in isoprene emissions (Alves et al., 2014, 2016). However,
even though these factors – solar radiation, temperature, and leaf phenology – have been
noted as important drivers of seasonal isoprene emissions, the way in which they control
seasonal emissions remains poorly represented in biogeochemical models.





In terms of modeling of isoprene emission from Amazonia, when light and
temperature are considered, MEGAN is a satisfactory tool for predicting short-term
changes in isoprene emissions (Karl et al., 2004, 2007). However, when long-term
changes are taking place, other factors, some still unknown, might be acting together,
which add uncertainties to isoprene seasonal emission estimates.
In this study, we present observations of seasonal variation of isoprene flux, solar
radiation, air temperature and canopy phenology from a primary rainforest site in central
Amazonia. The questions addressed are: (i) how much can seasonal isoprene fluxes be
explained by variations in solar radiation, temperature and leaf phenology, and (ii) how
can a consideration of leaf phenology observed in the field help to improve model
estimates of seasonal isoprene emissions. To this end, we correlate ground-based
isoprene flux measurements with environmental factors (light and temperature) and a
biological factor (leaf phenology). We compare seasonal ground-based isoprene flux
measurements to OMI satellite-derived isoprene flux. Lastly, we perform two simulations
with the MEGAN 2.1 to estimate isoprene fluxes: (1) with standard emission algorithms
and (2) with a modification in the leaf age algorithm derived from observed leaf
phenology.

**2. Material and methods**
**2.1. Site Description - Cuieiras Biological Reserve – K34 site**
Isoprene fluxes were measured at the 53 m K34 tower (2°36' 32.6" S, 60° 12'
33.4" W) on the Cuieiras Biological Reserve plateau, a primary rainforest reserve
approximately 60 km northwest of Manaus in Amazonas state, Brazil (Fig. 1). The K34



tower has been widely utilized for the past 15 years for a range of meteorological studies,
including energy and trace gas fluxes (de Araújo et al., 2010; Artaxo et al., 2013; Tóta et
al., 2012) and also tropospheric variables such as precipitable water vapor (Adams et al.,
2011, 2015). This reserve has an area of about 230 km² and is managed by the National
Institute for Amazonian Research (INPA). The site has a maximum altitude of 120 m and
the topography is characterized by 31% plateau, 26% slope and 43% valley (Rennó et al.,
2008). The vegetation in this area is considered mature, *terra firme* rainforest, and with
typical canopy height of 30 m with variation (20-45 m) throughout the reserve. More
details about soils and vegetation of this site are provided in Alves et al. (2016). Annual
precipitation is about 2500 mm and is dominated by deep atmospheric convection and
associated stratiform precipitation, December to May being the wet season and August to
September the dry season, when the monthly cumulative precipitation is less than 100
mm (Adams et al., 2013; Machado et al., 2004). Average air temperature ranges between
24 °C (in April) and 27 °C (in September) (Alves et al., 2016).

**2.2. Isoprene flux – Relaxed Eddy Accumulation system (REA)**
Isoprene flux measurements were conducted during intensive campaigns of five to
six days, during daytime (9:00-16:30, local time), from June 2013 to December 2013 at
the K34 tower. The REA system utilized for the isoprene flux measurements was
developed by the National Center for Atmospheric Research (NCAR)
NCAR/BEACHON REA Cassette Sampler), and has two basic components: 1) the main
REA box containing the adsorbent cartridges (stainless steel tubes filled with Tenax TA
and Carbograph 5 TD adsorbents) for up/down/neutral reservoirs, microcontroller,



battery, selection valves, and mass flow controller (200 ml min$^{-1}$) (MKS Instruments Inc.,
Model M100B01852CS1BV); and (2) a Sonic Anemometer (RM Young, Model
81000VRE) for high-rate wind velocity measurements (10 Hz). This REA system was
installed at a height of 48 m on the K34 tower (approximately 20 m above the mean
canopy height).

The technique segregated the sample flow according to sonic anemometer-derived

vertical wind velocity over the flux-averaging period (30 min). Isoprene fluxes ($F$) from
the REA system over this period were estimated from:
$$F = \overline{w'c'} = b\sigma_w(\overline{c_{up}} - \overline{c_{down}}) \tag{1}$$
where $b$ is an empirical proportionality coefficient (described below), $\sigma_w$ is the standard
deviation of $w$, and $\overline{c_{up}}$ and $\overline{c_{down}}$ are isoprene concentration averages in the up and
down reservoirs, respectively (Bowling et al., 1998). The $b$-coefficient was calculated
from the sonic temperature and heat flux by re-arranging the same equation, assuming
scalar similarity (Monin-Obukhov Similarity Theory):
$$b = \frac{\overline{w'T'}}{\sigma_w(T_{up} - T_{down})} \tag{2}$$

The REA sampler was operated with a "deadband" - a range of small $w'$ values,

centered on $\overline{w}$, over which the air was sampled through the "neutral" line. The deadband
used was $\pm 0.6\sigma_w$. The use of a deadband was advisable, because this increased the
differences in the measured concentrations ($\overline{c_{up}} - \overline{c_{down}}$) by sampling only larger eddies
(with larger concentration fluctuations) into the up/down reservoirs, reducing the
precision required for the analytical measurements. The $b$-coefficient was also computed
(from Eq. (2)) using the same deadband. For this study, the $b$-coefficient was calculated





for every 30 min. flux sampling period. The *b*-coefficient averaged 0.40 ± 0.06 and the
flux measurements were filtered for *b*-coefficients in the range of 0.3 to 0.6.

The air sampling was carried out with two tubing lines for up (+w') and down (-

w') and one tubing line for neutral sampling air (±0.6$\sigma_w$ - deadband), each consisting of
approximately 1.5 m long tubes (polytetrafluoroethylene, PTFE) positioned such that
they sampled air as close to the sonic anemometer as possible. Each inlet valve at the
main REA box prevented air from entering the inactive tube (up- in the case of down
sampling (-w') and down - in the case of up sampling (+w'), and both up and down in the
case of deadband), which otherwise would compromise the concentration differences
between up and down reservoirs and, consequently, the flux calculation.

The microcontroller recorded the sonic anemometer data and triggered the

segregation valves based on this data. The REA technique requires two initial data points
prior to each flux averaging period to be able to segregate the sample flow: (1) a mean
vertical wind velocity, $\bar{w}$ and (2) $\sigma_w$. The $\bar{w}$ determined the direction of the instantaneous
vertical wind velocity ($w' = w(t) - \bar{w}$) and $\sigma_w$ was required to calculate the deadband
threshold. Both the value of $\bar{w}$ and $\sigma_w$ were based on the values obtained from the last
flux-averaging period (30 min). The microcontroller stored all the necessary wind and
temperature information to compute all the parameters required in the equations (1) and
(2). More details on errors and uncertainties of the REA technique are found in section 1
(Supplementary Information).
**2.3. Isoprene concentrations**

The isoprene accumulated in the adsorbent cartridges was determined from

laboratory analysis. The tube samples were analyzed with a thermal desorption system





(TD) (Markes International, UK) interfaced with a gas chromatograph/flame ionization
detector (GC-FID) (19091J-413 series, Agilent Technologies, USA). After loading a tube
in the ULTRA Automatic Sampler (Model Ultra1, Markes International, UK), which was
connected to the thermal desorption system, the collected samples were dried by purging
for 5 minutes with 50 sccm of ultra-high purity helium (all flow vented out of the split
vent) before being transferred (300ºC for 10 min with 50 sccm of ultra-pure nitrogen) to
the thermal desorption cold trap held at -10 ºC (Unity Series1, Markes International, UK).
During GC injection, the trap was heated to 300°C for 3 min while back flushing with
carrier gas (helium) at a flow rate of 6.0 sccm directed into the column (Agilent HP-5 5%
Phenyl Methyl Siloxane Capillary 30.0 m X 320 µm X 0.25 µm). The oven ramp
temperature was programmed with an initial hold of 6 min at 27 °C followed by an
increase to 85 °C at 6 °C min$^{-1}$ followed by a hold at 200 °C for 6 min. The identification
of isoprene from samples was confirmed by comparison of retention time with a solution
of an authentic isoprene liquid standard in methanol (10 µg/ml in methanol, Sigma-
Aldrich, USA). The GC-FID was calibrated to isoprene by injecting 0.0, 23, 35, and 47
nL of the gas standard into separate tubes. The gas standard is 99.9% of 500 ppb of
isoprene in nitrogen (Apel & Riemer Environmental Inc., USA) and was injected into
separate tubes at 11 ml min$^{-1}$. The calibration curve (0.0, 23, 35, and 47 nL) was made
thrice before the analysis of the sample tubes of each campaign, with a mean correlation
coefficient equal to $R^2$=0.98. In addition, two standard tubes (with 35 nL of isoprene)
were run at every 20 sample tubes to check the system sensitivity. The limit of detection
of isoprene was equal to 48.4 ppt. All tube samples were analyzed as described above
with the exception of tube samples from June 2013 and July 2013. These were analyzed





in a TD/GC-MS-FID system from the Atmospheric Chemistry Division, NCAR (see
section 1 of supplementary information for more details).

Isoprene concentration was determined using the sample volume that was passed

through each tube. This volume was measured by integration of the mass flow meter
signal and stored within the REA data file. While sampling, the concentration found in
the blank tubes connected to the cartridge cassette in the REA box, but without flow, was
subtracted from the sample tube concentrations. The resulting concentration was used to
calculate isoprene flux (Eq. (1)) in mg m$^{-2}$ h$^{-1}$.

**2.4. Tower-camera derived leaf phenology and demography**

Upper canopy leaf phenology was monitored with Stardot RGB cameras (model

Netcam XL 3MP) installed at 53 m height on the K34 tower (Lopes et al., 2016; Wu et
al., 2016). The camera monitored forest on well-drained, infertile clay-rich soils of low
plateaus. Views were wide-angle and fixed, monitoring the same crowns over time and
excluding sky, so that auto-exposure was based only on the forest. Images were
automatically logged every two minutes from 09:00h to 12:30h, local time. Only images
acquired near local noon and under overcast sky (having even diffuse illumination) were
analyzed. Images were selected at six-day intervals. The camera monitored upper crown
surfaces of 53 living trees over 24 months (1 December 2011 to 31 November 2013).

We used a camera-based tree inventory approach to monitor leaf phenology at this

forest site (Lopes et al., 2016; Wu et al., 2016). Specifically, we tracked the temporal
trajectory of each tree crown, and assigned them into one of three classes: "leaf flushing"
(crowns which showed a large abrupt greening), "leaf abscising" (crowns which showed





large abrupt greying, which is the color of bare upper canopy branches) or "no change".
We then aggregated our census to the monthly scale to derive the monthly-average
percentages of trees with new leaf flushing and with old leaf abscission. The percentage
of tree crowns with green leaves (1 – the percentage of tree crowns with leaf abscission)
is termed as "green crown fraction" (Wu et al., 2016). We obtained a camera-based
canopy LAI by applying the same linear relationship between ground-measured LAI and
camera-derived green crown fraction, fitted at another central Amazon evergreen forest,
the Tapajós K67 tower site (Wu et al., 2016).

We also estimated the monthly canopy leaf demography by tracking the post-leaf-

flush age of each crown's leaf cohort and sorting them into three leaf age classes
throughout the year (young: <=2 months; mature: 3-5 months; and old: >=6 months)
(Nelson et al., 2014; Wu et al., 2016). By multiplying camera-derived total LAI by the
camera-derived fraction of crowns in a given age class, LAIs were derived for the three
leaf age classes: young leaf LAI, mature leaf LAI, and old leaf LAI. More details on
camera-derived LAI are in section 2 (Supplementary Information).

**2.5. Modeled isoprene flux estimates - MEGAN 2.1**

Isoprene fluxes measured by REA (K34 site) were compared with those estimated

by MEGAN 2.1. Isoprene emissions estimated by MEGAN 2.1 account for the main
processes driving variations in emissions (Guenther et al., 2012). The isoprene flux
activity factor for isoprene ($\gamma_i$) is proportional to emission response to light ($\gamma_P$),
temperature ($\gamma_T$), leaf age ($\gamma_A$), soil moisture ($\gamma_{SM}$), leaf area index (LAI) and $CO_2$
inhibition ($\gamma_{CO2}$) according to Eq. (3):



$$\gamma_i = C_{CE} LAI \gamma_P \gamma_T \gamma_A \gamma_{SM} \gamma_{CO_2} \qquad\qquad (3)$$
where $C_{CE}$ is the canopy environment coefficient. For this study, the canopy environment
model of Guenther et al. (2006) was used with a $C_{CE}$ of 0.57. MEGAN 2.1 was run
accounting for variations in light, temperature, and LAI. Based on changes in LAI, the
model estimated foliage leaf age. Both soil moisture and $CO_2$ inhibition activity factors
were set equal to a constant of 1, assuming these parameters do not vary. Details on
model settings are found in Guenther et al. (2012).
Photosynthetic photon flux density (PPFD) and air temperature inputs for all
model simulations were obtained from measurements at K34 tower. PPFD and air
temperature measured at tower top, every 30 minutes, were hourly averaged. Data gaps
during certain months occurred in 2013, but at least 15 days of hourly average PPFD and
air temperature were obtained for model input. LAI inputs were acquired from the
Moderate Resolution Imaging Spectroradiometer (MODIS) satellite observations for the
same period of the isoprene flux measurements. The level-4 LAI product is composited
every 8 days at 1-km resolution on a sinusoidal grid (MCD15A2H) (Myneni, 2015).
Additionally, by comparison with the standard MEGAN 2.1 model that uses MODIS-
derived LAI variation, here we also used LAI fractionated into different leaf ages, which
were obtained from tower camera observations (as described in the section above). The
number of data inputs to the MEGAN simulations is summarized in table 1.

**2.6. Satellite-derived isoprene flux estimates**
Top-down isoprene emission estimates over the 0.5 degree region around the tower were
obtained by applying a grid-based source inversion scheme (Stavrakou et al., 2009, 2015)



constrained by satellite formaldehyde (HCHO) columns, measured in the UV-visible by
the Ozone Monitoring Instrument (OMI) onboard the Aura satellite launched in 2004.
HCHO is a high yield intermediate product in the isoprene degradation process
(Stavrakou et al., 2014). The source inversion was performed using the global chemistry-
transport model IMAGESv2 (Intermediate Model of Annual and Global Evolution of
Species) at a resolution of 2º × 2.5º and 40 vertical levels from the surface to the lower
stratosphere (Stavrakou et al., 2014, 2015). The a priori isoprene emission inventory was
taken from MEGAN-MOHYCAN (Stavrakou et al., 2014, http://emissions.aeronomie.be,
Bauwens et al. 2017). Given that the OMI overpass time is in the early afternoon (13:30,
local time), and the mostly delayed production of formaldehyde from isoprene oxidation,
the top-down emission estimates rely on the ability of MEGAN to simulate the diurnal
isoprene emission cycle and on the parameterization of chemical and physical processes
affecting isoprene and its degradation products in IMAGESv2. For this study, we use
daily (24 hours), mean satellite-derived isoprene emissions derived from January 2005 to
December 2013. More details can be found in Stavrakou et al. (2009,  2015) and
Bauwens et al. (2016).

**3. Results**

The experimental site of this study showed seasonal variation in air temperature

and in photosynthetic active radiation (PAR) (Fig. 2a,b) that was comparable to the
seasonality presented by the OMI satellite-derived isoprene fluxes for the K34 site
domain (Fig. 2c). The interannual variation in the seasonality of these environmental
factors, air temperature and PAR, was correlated to the one presented by the satellite-



derived isoprene fluxes, with the highest correlation found between satellite-derived
isoprene fluxes and air temperature. Isoprene fluxes and PAR - $R^2$ ranged from 0.34 to
0.83 $p<0.05$; isoprene fluxes and air temperature - $R^2$ ranged from 0.61 to 0.91, $p<0.01$,
from 2005 to 2013. Maxima and minima of PAR, air temperature, and satellite-derived
isoprene fluxes were observed during the dry and the dry-to-wet transition seasons, and
the wet and the wet-to-dry transition seasons, respectively.

As opposed to the average (2005-2013) flux peaking in September, the 2013

results suggest a maximum in October, and are found to be substantially lower during the
2013 dry season compared to the average of the dry season estimates (reduction of ~31%)
(Fig. 2c). The timing of the maximum is not supported by the ground-based observations,
peaking in September, but the magnitude of flux estimates in these two months are in
good agreement. In the wet-to-dry transition period, the small reduction in satellite-based
isoprene fluxes in July 2013, compared to the neighboring months, is corroborated by a
similar behavior in the ground-based isoprene fluxes (Fig. 3d). However, the drop in the
observations is much stronger than in the top-down estimates (factor of 3 vs. a 70%
difference).

Different from satellite-derived fluxes, ground-based isoprene fluxes measured

with the REA system have not shown significant correlation with PAR and air
temperature for the year 2013 (Table 2 and Fig. 3). Ground-based isoprene fluxes also
showed the maximum emission during the dry season (September), but emissions
remained high in the beginning of the wet season (December), which was not observed in
the seasonal behavior of PAR and air temperature. When averages of air temperature and
PAR measured only during the same days of REA isoprene flux measurements were



compared to isoprene fluxes, the correlations coefficients increased, but were still not
statically significant (Table 2).

The forest leaf quantity, shown as Leaf Area Index (LAI), varied little over the

year when the total LAI was examined. However, when total LAI was fractionated into
three different leaf age classes – young LAI (<=2 months), mature LAI (3-5 months), and
old LAI (>=6 months), seasonal variation of each age class appears (Fig. 4). To
understand how those LAI age fractions are related to the isoprene seasonality, ground-
based fluxes of this compound were compared to the LAI age fractions estimated over the
entire year (Fig. 4). The highest emissions were observed when the number of trees with
mature leaves (mature LAI) was increasing and the number of trees with old leaves (old
LAI) was decreasing. Considering seasonal changes in PAR, air temperature, and mature
LAI, the latter presented the highest correlation coefficient, explaining 59% of the
seasonal isoprene emission variations (Table 2).

Isoprene flux simulations carried out with MEGAN 2.1 reveal similarities with the

magnitudes observed during several months. But, MEGAN 2.1 did not fully capture the
observed seasonal behavior (Fig. 5). Even though the leaf age algorithm of MEGAN 2.1
was parameterized with local leaf phenology observations, giving the highest correlation
coefficient with observed fluxes (Table 2), isoprene flux simulations with local
CAMERA-LAI inputs showed only a reduction in isoprene flux magnitudes. The
seasonal behavior observed was the same as in the estimates from the default MEGAN
2.1 with MODIS-LAI inputs. Regressions between averages of observations and
MEGAN 2.1 estimates, with CAMERA-LAI and MODIS-LAI inputs, were weak and not
statically significant (Table 2).



As a sensitivity test, observations of isoprene emission capacity at different leaf

ages of a central Amazonian hyper-dominant tree species, *Eschweilera coriacea* (Alves et
al., 2014), were used to parameterize the MEGAN 2.1 leaf age algorithm. Leaf level
measurements of isoprene emission capacity are scarce in Amazonia. To the authors'
knowledge, Alves et al. (2014) is the only available data of leaf level isoprene emission
capacity at different leaf ages of a central Amazonian tree species, which were therefore
used for the MEGAN sensitivity test.

Further simulations were performed with modifications in the leaf age emission

activity factor (EAF), which is dimensionless and is defined as the emission relative to
the emission of mature leaves that are, by definition, set equal to one. A new EAF was
assigned for each age class, based on observations of emissions of *E. coriacea* (Fig. 6).
Leaf age fraction distribution was provided with input of LAI from MODIS (MODIS-
LAI) and from LAI-derived field observations (CAMERA-LAI) (Fig. 4). The simulation
with the leaf age algorithm parameterized for EAF changes and with MODIS-LAI was
similar to the one without changes in the EAF (MEGAN 2.1 default). The simulation
with leaf age algorithm parameterized with changes in the EAF and with CAMERA-LAI
inputs showed reduced emissions, but a seasonal curve closer to that of isoprene flux
observed at K34 ($R^2 = 0.52$, $p<0.05$) (Table 2).

**4. Discussion**

This study addressed two main questions with respect to the seasonality of

isoprene fluxes in central Amazonia and identified possible limitations in our current
understanding related to these questions.



### 4.1. How much can seasonal isoprene fluxes be explained by variations in solar radiation, temperature, and leaf phenology?

Our finding that isoprene emissions are higher during the warmer season is consistent with previous findings that emissions from tropical tree species are light dependent and stimulated by high temperatures (Alves et al., 2014; Harley et al., 2004; Jardine et al., 2014; Kuhn et al., 2002, 2004a, 2004b). Indeed, satellite-derived isoprene fluxes (2005-2013 years) were well correlated to PAR and even more to air temperature for all years. However, high ground-based isoprene emissions were observed until late of dry-to-wet transition season, when mean PAR and air temperature were already decreasing.

The reasons why satellite-derived isoprene fluxes are weakly correlated to ground-based isoprene fluxes can be attributed to either the difference in the studied scales (e.g. local effects could have major influences on ground-based isoprene fluxes) and/or the uncertainties associated with the methodologies used to estimate or calculate fluxes. The high correlation between satellite-based fluxes and air temperature or PAR is not unexpected, because higher temperatures and solar radiation fluxes favor isoprene emissions. Note however that the satellite-derived fluxes might also be subject to inherent uncertainties, due to the existence of other HCHO sources, in particular biomass burning (during the dry season) and methane oxidation. Since these latter contributions are favored by high temperature and radiation levels, they could possibly contribute to the high correlation found between satellite-based isoprene and meteorological variabales.

For the ground-based emission, isoprene fluxes were determined by REA measurements that were carried out for six days per month. Therefore, the low correlation





between ground-based isoprene fluxes and air temperature and PAR could partially result
from limited qualified data.

Another factor correlated to ground-based isoprene fluxes is the leaf phenology

(in this study, LAI fractionated into age classes). The variation of mature LAI correlated
better to ground-based isoprene fluxes than to other factors (K34 site – $R^2$=59%, $p$<0.05),
suggesting that the increasing isoprene emissions could partially follow the increasing of
mature leaves (Fig. 4). Wu et al. (2016) suggested that leaf demography (canopy leaf age
composition) and leaf ontogeny (age-dependent photosynthetic efficiency) are the main
reasons for the seasonal variation of the ecosystem photosynthetic capacity in Amazonia.
Since photosynthesis supplies the carbon to the methyl erythritol phosphate pathway to
produce isoprene (Delwiche and Sharkey, 1993; Harley et al., 1999; Lichtenthaler et al.,
1997; Loreto and Sharkey, 1993; Rohmer, 2008; Schwender et al., 1997), and as isoprene
emissions are strongly dependent on leaf age and mainly emitted by mature leaves (Alves
et al., 2014), seasonal changes in the forest leaf-age fractions may also influence the
seasonality of isoprene emissions, suggesting higher emissions in the presence of more
mature leaves and during high ecosystem photosynthetic capacity efficiency.

Understanding the correlations among light, temperature, leaf phenology (LAI

fractionated into age classes), and isoprene is not straightforward. The weak correlation
of seasonal changes between isoprene and light and temperature might be due to seasonal
changes in the isoprene dependency to environmental factors and biological factors. Light
and temperature peaked at the dry season; mature LAI, Gross Primary Productivity (GPP)
and photosynthetic capacity peaked at the wet season (Wu et al., 2016); and ground-
based isoprene fluxes were high from the end of the dry to the dry-to-wet transition



seasons. This might suggest that isoprene emissions are stimulated by light and high

temperature during the beginning of the dry season and offset by the lower amount of

mature leaves. During the wet season, isoprene emissions could be stimulated by the

higher abundance of mature leaves and offset by the lower light availability and lower

temperature. But, at the end of the dry and at dry-to-wet transition seasons, there is a

combination of high light and high temperature with high amount of mature leaves,

possibly favoring high isoprene emissions.

This is supported by findings of a temperate plant species showing that LAI

dependency (changes in leaf age) was the most important factor affecting isoprene

emission capacity, but when LAI decreased, and senescence started at the end of the

summer, the isoprene dependency to PAR and air temperature was as high as the period

when PAR and air temperature reached their maximum (Brilli et al., 2016). This shows

seasonal variation in the strength of dependency to each factor that affects emissions.

Furthermore, we demonstrate a lack of a general correlation between ecosystem

seasonal cycles of photosynthetic capacity or GPP and isoprene emissions (Table 2). This

is consistent with previous studies that provide evidence that alternative non-

photosynthetic pathways may contribute to isoprene synthesis under stress (Loreto and

Delfine, 2000), which may then lead to a decoupling of isoprene emission from

photosynthesis at high temperatures (Foster et al., 2014). In this light, it could be

suggested that the strong correlation between GPP and isoprene emission during leaf

phenology (Kuhn et al., 2004a) is reduced during conditions of high temperature.

As discussed above, separating the effects of changing temperature and light from

leaf phenology in canopy isoprene fluxes could allow for a more accurate quantification



and for a better understanding of seasonal isoprene flux. Here, we indicate that leaf
phenology plays an important role in seasonal variation of isoprene emissions, especially
because different leaf ages present different isoprene emission capacity and the
proportion of leaf age changes seasonally in Amazonia. However, when air temperature
is the highest, isoprene emission could be more stimulated by this factor, even though
mature LAI is still not at its maximum. We suggest future research to verify whether tree
species that present a regular seasonal leaf flushing are isoprene emitters and the strength
of those emissions by leaf age.

**4.2. How can a consideration of leaf phenology observed in the field help to improve**
**model estimates of seasonal isoprene emissions?**
Modeling of isoprene emissions from the Amazonian rainforest has been carried
out for around thirty years. The first models were simplified and parameterized with
observations from a few short field campaigns (see Table 1 of Alves et al., 2016). With
the increase in available data, more driving forces of isoprene emission were accounted
for in the latest versions of models, as the case of the MEGAN 2.1, which has been
improved with a multi-layer canopy model that accounts for light interception and leaf
temperature within the canopy, and includes changes in emissions due to leaf age that are
typically driven by satellite retrievals of LAI development (Guenther et al., 2012).
Results presented here are from MEGAN 2.1 estimates with local observations of
PAR, air temperature, and satellite-based leaf phenology. Initially, the default MEGAN
2.1 simulations did not fully capture the seasonal pattern of observed isoprene emission,
with none-significant correlation between model estimates and observations ($R^2$= 0.16,



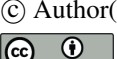

*P*>0.05, Table 2). This could be due to the near saturation of LAI seasonality in
Amazonian evergreen forests and poor representation of leaf age effect on isoprene
emission capacity of tropical tree species in the default MEGAN 2.1. Further, by using
the camera-derived LAI phenology and the leaf age demographics to update the leaf age
algorithm of the default MEGAN 2.1, we improved estimates of the proportion of leaves
in different leaf age categories for the site, but there were a lack of observations for
assigning the relative isoprene emission capacity for each age class.

It has been suggested that MEGAN uncertainties are mostly related to short-term

and long-term seasonality of the isoprene emission capacity (Niinemets et al., 2010). For
instance, for an Asian tropical forest, isoprene emission capacity was reported to be four
times lower than the default value of the MEGAN model (Langford et al., 2010), whereas
aircraft flux measurements in the Amazon were 35% higher than the MEGAN values (Gu
et al., 2017); and satellite retrievals suggested significantly lower isoprene emissions (30-
40 % in Amazonia and northern Africa) with respect to the MEGAN-MOHYCAN
database (Bauwens et al., 2016). These all demonstrate that isoprene emission capacity is
not well represented in the model for regions where there are few or no measurements.

With a sensitivity test, we parameterized the MEGAN 2.1 leaf age algorithm with

observed isoprene emission capacity among different leaf ages of *E. coriacea* (Alves et
al., 2014). The resulting simulation showed that by knowing the leaf age class
distribution and the isoprene emission capacity for each age class, MEGAN 2.1 estimates
can be improved and better agree with observations in terms of seasonal behavior. To
date, there is very little information about isoprene emission capacity for different leaf
ages of Amazonian plant species (Alves et al., 2014; Kuhn et al., 2004a). The scarcity of



observational studies in the field, along with the huge biodiversity and heterogeneity of
the Amazonian ecosystems, creates a challenge to optimize the isoprene emission
capacity parameterization in MEGAN and other models. Therefore, while introducing
local seasonal changes of canopy leaf age fractions in the model should improve
estimates, seasonal variations in isoprene emission capacity also need to be characterized
to better represent the effects of leaf phenology on ecosystem isoprene emissions.

**4.3. Possible limitations**

This study correlates available data of different scales and approaches. Thus, there

are limitations that need to be considered. One is the uncertainty related to the method
used to measure ground-based isoprene fluxes. The uncertainties of the REA flux
measurements ranged from 27.1% to 44.9% (more details in section 1 of Supplementary
Information). However, this study shows the largest dataset of seasonal isoprene fluxes in
Amazonia presented to date and results presented here are similar to previous
investigations, when same seasons are compared (see Table 1 of Alves et al., 2016).

Another limitation is the uncertainty of MEGAN estimates. It has been shown that

models tend to agree with observations within ~30% for canopy scale studies with site-
specific parameters (Lamb et al., 1996). Here, part of the weak correlation between
observations and MEGAN 2.1 estimates is possibly due to short periods of measurements
and data gaps. There were data gaps of PAR and temperature for a few months in 2013.
This could influence the mean flux obtained from model estimates. Also, REA
measurements were carried out in intensive campaigns of six days per month, which may



not represent the flux for the entire month. Therefore, the limited data availability is still
challenging our understanding of isoprene emission seasonality.

**5. Summary and Conclusions**

To understand the pattern of isoprene seasonal fluxes in Amazonia is a difficult

task when considering the important role of Amazonian forests in accounting for global
BVOC and very limited field based observations in Amazonia. Seasonal variation of light
and temperature are thought to primarily drive isoprene seasonal emissions. However,
less notable factors might also influence ecosystem isoprene emission. Here, we suggest
that leaf phenology, especially when accounting for the effect of leaf demography
(canopy leaf age composition) and leaf ontogeny (age-dependent isoprene emission
capacity), has an important effect on seasonal changes of the ecosystem isoprene
emissions, which could play even more important role in regulating ecosystem isoprene
fluxes than light and temperature at seasonal timescale.

Albeit there are uncertainties related to measurements and modeling, results

presented here suggested that the unknown isoprene emission capacity for the different
leaf age classes found in the forest may be the main reason why MEGAN 2.1 did not
represent well the observed seasonality of isoprene fluxes. Additionally, part of these
model uncertainties arises because of a lack of representations of canopy structure and
light interception, including within-canopy variation in leaf functional traits; the leaf
phenology within the canopy; the physical processes by which isoprene is transported
within and above the forest canopy; chemical reactions that can take place within the
canopy; and, the most difficult to assess, emission variation due to the huge biodiversity



in Amazonia. Therefore, more detailed measurements of source and sink processes are
encouraged to improve our understanding of the seasonality of isoprene emissions in
Amazonia, which will improve surface emission models and will subsequently lead to a
better predictive vision of atmospheric chemistry, biogeochemical cycles, and climate.

**6. Data Availability**
Even though the data are still not available in any public repository, the data are
available upon request from the main author.

**7. Acknowledgements**
The authors thank the National Institute for Amazonian Research (INPA) for continuous
support. We acknowledge the support by the Large Program of Biosphere-Atmosphere
Interactions (LBA) for the logistics and the micrometeorological group for their
collaboration concerning the meteorological parameters. We acknowledge Kolby Jardine
for providing the gas standard to calibrate the analytical system, and Paula Regina Corain
Lopes for the help in the fieldwork. J.W. is supported by DOE BER funded NGEE-
Tropics project (contract # DE- SC00112704) to Brookhaven National Laboratory.

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

**Tables**
**Table 1: Environmental and biological factors used to input the MEGAN 2.1: number of**
**days with data available for each variable for the year 2013**

| | Jan | Feb | Mar | Apr | May | Jun | Jul | Aug | Sep | Oct | Nov | Dec |
|---|---|---|---|---|---|---|---|---|---|---|---|---|
| **PAR** | $n$=31 | $n$=28 | $n$=31 | $n$=30 | $n$=31 | $n$=30 | $n$=31 | $n$=15 | $n$=30 | $n$=18 | $n$=19 | $n$=15 |
| **Air temperature** | $n$=31 | $n$=28 | $n$=31 | $n$=30 | $n$=31 | $n$=30 | $n$=31 | $n$=15 | $n$=30 | $n$=18 | $n$=19 | $n$=15 |
| **CAMERA-LAI*** | $n$=5 | $n$=4 | $n$=5 | $n$=5 | $n$=5 | $n$=5 | $n$=5 | $n$=5 | $n$=5 | $n$=5 | $n$=5 | $n$=5 |
| **MODIS-LAI**** | $n$=4 | $n$=4 | $n$=4 | $n$=3 | $n$=5 | $n$=4 | $n$=4 | $n$=4 | $n$=4 | $n$=3 | $n$=4 | $n$=4 |

* Number of days with images analyzed to derive CAMERA-LAI as described in section 2.4.
** Number of days that the satellite passed over the site domain.









**Table 2: Correlation coefficient, *R*, of regressions for ground-based isoprene flux,**
**satellite-derived isoprene flux, environmental factors, biological factors, and**
**MEGAN 2.1 simulations**

| | Ground-based isoprene flux | Satellite-derived isoprene flux (2013 year) |
|---|---|---|
| **PAR** | 0.007[a] | 0.55[c] |
| **PAR – REA measurement days** | 0.11[a] | ----- |
| **Air temperature** | 0.15[a] | 0.79[c] |
| **Air temperature – REA measurement days** | 0.39[a] | ----- |
| **young LAI** | 0.04[a] | 0.35[b] |
| **mature LAI** | 0.59[b] | 0.05[a] |
| **old LAI** | -0.6[b] | -0.4[b] |
| **Photosynthetic capacity\*** | 0.49[a] | ----- |
| **GPP\*** | 0.36[a] | ----- |
| **MEGAN (MODIS-LAI)** | 0.16[a] | 0.76[c] |
| **MEGAN (CAMERA-LAI)** | 0.11[a] | 0.67[c] |
| **MEGAN (MODIS-LAI) EAF changed** | 0.19[a] | 0.66[c] |
| **MEGAN (CAMERA-LAI) EAF changed** | 0.52[b] | 0.59[c] |
| **Ground-based isoprene flux** | ----- | 0.13[a] |

PAR, photosynthetic active radiation; GPP, gross primary productivity;
EAF, emission activity factor;
\* Data from Wu *et al.* (2016)
[a] not statistically significant (*P > 0.05*)
[b] statistically significant (*P < 0.05*)
[c] statistically significant (*P < 0.001*)

**Figure captions**
**Figure 1**. Location of the experimental site in central Amazonia – K34 tower. Hill-
shaded digital elevation data used as background topography is from the Shuttle Radar
Topography Mission, with resolutions of ~900m (top panel) and ~30m (lower panel).
White ring indicates two km radius around the flux tower. Elevation scale for lower panel
is "meters above sea level".





**Figure 2**. Monthly averages of photosynthetic active radiation (PAR) (a) and air
temperature (b) from 2005 to 2013 at the K34 tower site (measured every 30 min during -
6:00-18:00h, local time). OMI satellite-derived isoprene flux in a resolution of 0.5º
centered on K34 tower site from 2005 to 2013 (c). Monthly averages of isoprene flux
were scaled to 10:00-14:00, local time. Error bars represent one standard error of the
mean.
**Figure 3**. Monthly cumulative precipitation given by the Tropical Rainfall Measuring
Mission (TRMM) for the K34 tower domain in 2013 (a) Monthly averages of PAR (b)
and air temperature (c), both measured every 30 minutes during 6:00-18:00h, local time,
at the K34 tower site in 2013. Isoprene flux measured with the REA system at the K34
tower site in 2013 (d).
**Figure 4.** CAMERA-LAI derived for the K34 tower site. CAMERA-LAI data are
presented in three different leaf age classes: young LAI, mature LAI and old LAI. Error
bars represent one standard deviation from the mean. Background color shadings indicate
each season and are explicit in the legend. DWT season and WDT season stand for the
dry-to-wet transition season and the wet-to-dry transition season, respectively.
**Figure 5**: Isoprene flux observed (REA) and estimated with MEGAN 2.1 default mode,
leaf age algorithm driven by MODIS-LAI, and with MEGAN 2.1 leaf age algorithm
driven by CAMERA-LAI. EAF stands for emission activity factor, which was changed
for the different leaf age classes based on emissions of *E. coriacea* (Alves et al., 2014).
**Figure 6**. Emission activity factor (EAF) of isoprene for each leaf age class assigned in
the default mode of MEGAN 2.1 proportional to leaf age class distribution derived from
field observations (CAMERA-LAI) (a) Isoprene EAF for each leaf age class, obtained



from leaf level measurements of the tree species *E. coriacea,* proportional to leaf age
class distribution derived from field observations (CAMERA-LAI) (b) Observations of
the tree species *E. coriacea* (Alves *et al.,* 2014) and CAMERA-LAI are both from the
K34 site.

**Figures**

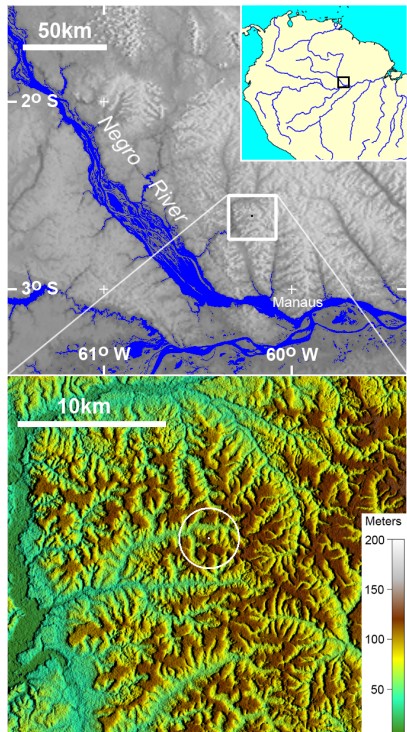


**Figure 1**. Location of the experimental site in central Amazonia – K34 tower. Hill-
shaded digital elevation data used as background topography is from the Shuttle Radar
Topography Mission, with resolutions of ~900m (top panel) and ~30m (lower panel).
White ring indicates two km radius around the flux tower. Elevation scale for lower panel
is "meters above sea level".



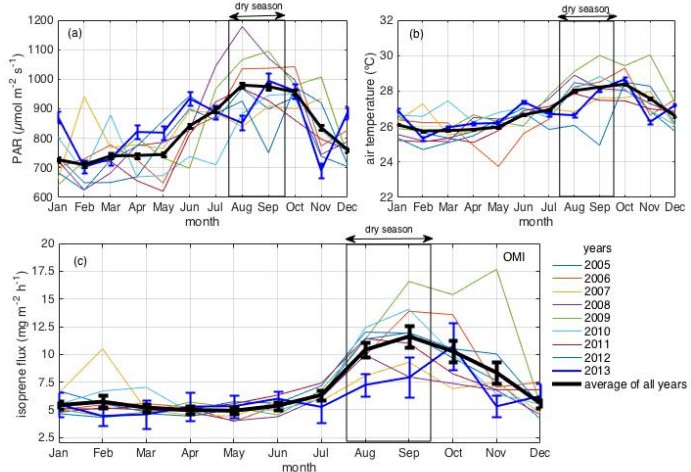


**Figure 2**. Monthly averages of photosynthetic active radiation (PAR) (a) and air
temperature (b) from 2005 to 2013 at the K34 tower site (measured every 30 min during -
6:00-18:00h, local time). OMI satellite-derived isoprene flux in a resolution of 0.5º
centered on K34 tower site from 2005 to 2013 (c). Monthly averages of isoprene flux
were scaled to 10:00-14:00, local time. Error bars represent one standard error of the
mean.







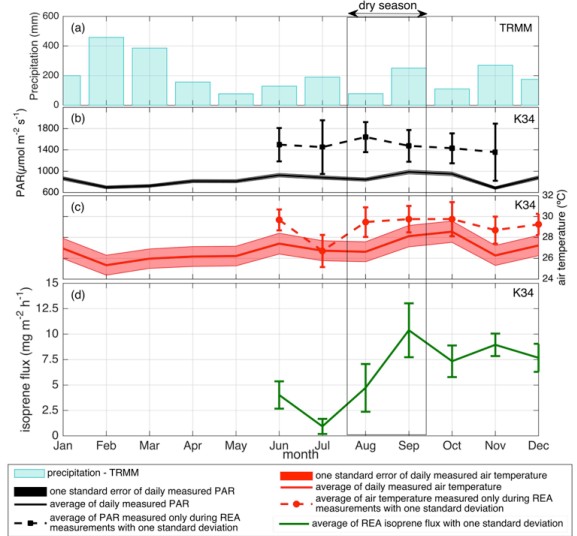

**Figure 3**. Monthly cumulative precipitation given by the Tropical Rainfall Measuring
Mission (TRMM) for the K34 tower domain in 2013 (a) Monthly averages of PAR (b)
and air temperature (c), both measured every 30 minutes during 6:00-18:00h, local time,
at the K34 tower site in 2013. Isoprene flux measured with the REA system at the K34
tower site in 2013 (d).






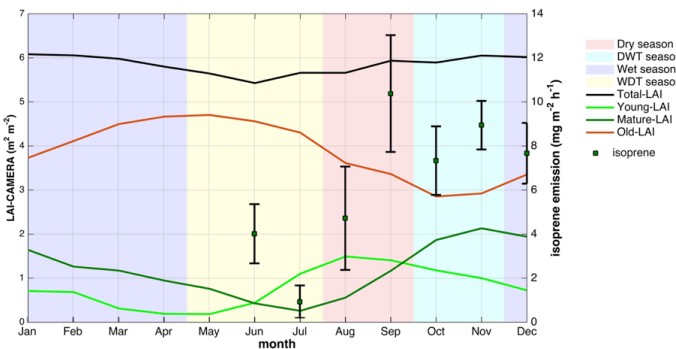


**Figure 4.** CAMERA-LAI derived for the K34 tower site. CAMERA-LAI data are

presented in three different leaf age classes: young LAI, mature LAI and old LAI. Error

bars represent one standard deviation from the mean. Background color shadings indicate

each season and are explicit in the legend. DWT season and WDT season stand for the

dry-to-wet transition season and the wet-to-dry transition season, respectively.

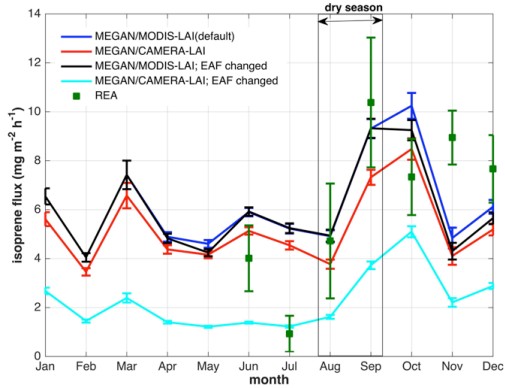

890

**Figure 5**: Isoprene flux observed (REA) and estimated with MEGAN 2.1 default mode,

leaf age algorithm driven by MODIS-LAI, and with MEGAN 2.1 leaf age algorithm

driven by CAMERA-LAI. EAF stands for emission activity factor, which was changed

for the different leaf age classes based on emissions of *E. coriacea* (Alves et al., 2014).





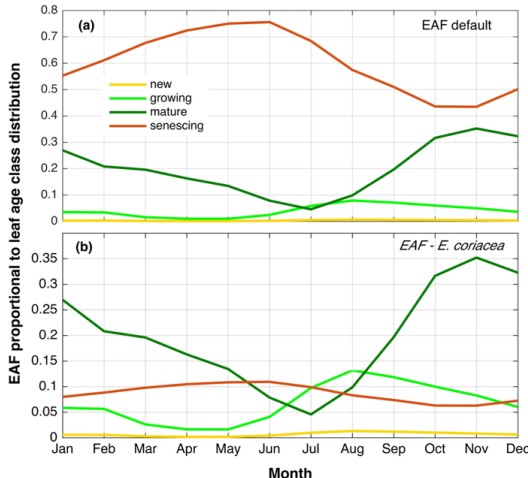

895

**Figure 6**. Emission activity factor (EAF) of isoprene for each leaf age class assigned in

the default mode of MEGAN 2.1 proportional to leaf age class distribution derived from

field observations (CAMERA-LAI) (a) Isoprene EAF for each leaf age class, obtained

from leaf level measurements of the tree species *E. coriacea*, proportional to leaf age

class distribution derived from field observations (CAMERA-LAI) (b) Observations of

the tree species *E. coriacea* (Alves *et al*., 2014) and CAMERA-LAI are both from the

K34 site.

903
904
905