# Peer review of "Leaf phenology as one important driver of seasonal changes in isoprene emission in"

_Biogeosciences, 2018_

## Referee Comment (RC1) · Anonymous Referee #1 · 29 Mar 2018

The study by Alves et al. provides a context for phenological control over isoprene emissions in an Amazonian tropical forest. I think that the results are interesting, but only because this is a tropical forest. The potential for phenology and leaf age to control isoprene emission rates has been recognized in past studies going back 25 years. Studies by Fall, Monson, Harley, Litvak, Sharkey, Loreto and many others have clearly shown these relationships in temperate forest trees. The Alves et al. study is most interesting because it deals with a tropical forest, for which this type of insight is missing.

The main critique I level against the study is that it is written to largely ignore this

past body of work, and the broader context of phenological influences over isoprene emission, and instead makes it sound like this is a new relationship discovered since 2013.

I recommend a major revision of the work that honestly takes into account the historical context of the phenology-emission relationship and its relevance to the observations made in the tropical forest. In this revision I recommend making it clear that the novel aspects of the current work are that it (1) is among the first to show the importance of the phenology-emission relationship in a tropical forest, and (2) that it allows for the MEGAN model to be modified to better predict emissions in tropical forests.

Lines 66-68. The phrase "as BVOC emissions are regarded as highly significant for ecosystem productivity (Kesselmeier et al., 2002) with isoprene being the most emitted hydrocarbon, it thereby plays an important role in carbon balance", is worded strangely. What does "highly significant for ecosystem productivity" mean? Why is that part of the phrase supported with a reference, but the next part of the phrase, "it thereby plays an important role in carbon balance", is not supported by a reference? Does the Kesselmeier reference cover both parts of the phrase? If not, it seems that a second reference is needed for the second part of the phrase. I am especially interested in what is meant by "important role", as my understanding is that isoprene emission occurs as a small absolute flux compared to overall carbon fluxes (e.g., approximately 1000 times lower).

Lines 402-406. The phrase "and as isoprene emissions are strongly dependent on leaf age and mainly emitted by mature leaves (Alves et al., 2014), seasonal changes in the forest leaf-age fractions may also influence the seasonality of isoprene emissions, suggesting higher emissions in the presence of more mature leaves and during high ecosystem photosynthetic capacity efficiency." I found this to be a bit of an egregious claim by the authors. The implication is that the dependence of isoprene emissions on leaf age and phenology was only discovered by the authors in 2014 ignores a rich literature that has shown the effects of leaf age and phenology on isoprene emissions

going back at least two decades. There are many past studies showing, in explicit terms, the effects of leaf age and phenology on isoprene emissions. The authors seem gracious in citing the rich literature connecting photosynthate to isoprene emissions, but then take sole credit for discovering the connection between phenology and isoprene emissions. This should be corrected so that the true scope of the problem and related past research is brought honestly into this paper.

Lines 428-432. The phrase "This is consistent with previous studies that provide evidence that alternative non-photosynthetic pathways may contribute to isoprene synthesis under stress (Loreto and Delfine, 2000), which may then lead to a decoupling of isoprene emission from 432 photosynthesis at high temperatures (Foster et al., 2014)." This seems to be a rather large and speculative jump in logic. There is no reason to suspect that the seasonal offset between GPP and isoprene emission rate is due to the use of stored carbon sources. In fact, the past literature (going back into the 1990s) shows that the leaf age effect is likely due to developmental (ontogenetic) patterns of isoprene synthase activity. Thus, the phenological constraint over isoprene emission (1) has the potential to override the correlation between photosynthesis rate and isoprene emission rate, and (2) this is due to an enzymatic limitation, not a limitation of carbohydrate availability. The authors seem to be unaware of this past literature as it is not mentioned in their paper. This should be corrected.

Lines 512-513. The phrase "However, less notable factors might also influence ecosystem isoprene emission." Once again, this phrase makes it seem like very few past studies have considered factors like phenology or leaf age as an important control over isoprene emissions. Actually, these factors have been recognized as just as important as temperature and light for over 25 years. The authors need to present their results in a way that embeds them honestly within the rich past tradition of isoprene emissions research.

---

## Referee Comment (RC2) · Anonymous Referee #2 · 24 Apr 2018

Alves et al. present a 7-month observation of isoprene flux in central Amazonia, and demonstrate the role of leaf age in controlling its seasonal variation. This study deserves documentation because it provides a long observational record of isoprene emissions and in-situ co-monitored leaf phenology, which is scarce in the tropics. However, I agree with the other reviewer that this manuscript ignores a pool of previous literature. Some other issues may need to be addressed as well before accepted for publication.

L64-68: What is the contribution of isoprene to total $CO_2$ emission in percentage? To my knowledge the number is very small.

[Figure]

L81: "drivers of isoprene" should be "drivers of isoprene emissions".

L83-88: "canopy phenology could therefore be an important seasonal driver. . ." Does the phenological control on isoprene emissions only occur through photosynthesis? Kuzma and Fall, 1993 suggested that the enzyme activity regulates the isoprene emission in response to leaf development. The authors may want to replace the sentence with a paragraph of literature review (including mid-latitude studies) on the theory and observations of isoprene emission versus leaf phenology. See review paper Harrison et al., 2013 (Table S2) and many others, Niinemets, Monson, Sharkey, etc.

L132: How did you choose the 5 or 6 days every month for measurement? What are the cloud conditions?

L305-314: In 2013, the monthly variation of satellite-derived isoprene emission is totally wrong compared to in-situ measurement. Is it because you only have a few days' measurement each month? I suggest to include the 2013 satellite isoprene curve in Figure 3 for a direct comparison. Include both monthly average and the REA period average.

L333: Does Table 2 show R or Rˆ2? "Explaining 59% of variations" usually refers to Rˆ2 values. The abstract should be consistent, too.

L342: "Regression" should be "Correlation".

L352-362, Figure 6: No matter with EAF changes or not, the MEGAN monthly variations look more similar to the satellite-derived isoprene emissions. Again, is this because the in-situ observation only includes a few days every month? I wonder whether these days can represent emissions during the whole month. Is MEGAN run at a day-by-day basis? If so, the authors may try take out the MEGAN simulations during the same days as the REA measurement to see whether the correlations are improved.

Another possibility is soil-moisture dependence. Quite a few studies showed the importance of water availability, e.g. Pegoraro et al., 2004-2006, Zheng et al., 2015, 2017,

etc. In Figure 3, observed isoprene flux shows a similar monthly pattern as the TRMM precipitation in dry and dry-to-wet seasons (when water is limited). The authors may want to do a MEGAN sensitivity test that includes soil moisture dependence or at least discuss the role of soil moisture in Section 4.1.

L434: The wording "during leaf phenology" is strange.

Figure 2, 3, 6: As a convention, the panel numbers (a)(b)(c) are usually placed in front of description.

Some references: Kuzma, Jennifer, and Ray Fall. "Leaf isoprene emission rate is dependent on leaf development and the level of isoprene synthase." Plant physiology 101.2 (1993): 435-440. Harrison, Sandy P., et al. "Volatile isoprenoid emissions from plastid to planet." New Phytologist 197.1 (2013): 49-57. Niinemets, Ülo, et al. "The emission factor of volatile isoprenoids: stress, acclimation, and developmental responses." Biogeosciences 7.7 (2010): 2203. Pegoraro, E., et al. "Effect of drought on isoprene emission rates from leaves of Quercus virginiana Mill." Atmospheric Environment 38.36 (2004): 6149-6156. Pegoraro, Emiliano, et al. "The interacting effects of elevated atmospheric $CO_2$ concentration, drought and leaf-to-air vapour pressure deficit on ecosystem isoprene fluxes." Oecologia 146.1 (2005): 120-129. Pegoraro, Emiliano, et al. "Drought effect on isoprene production and consumption in Biosphere 2 tropical rainforest." Global Change Biology 12.3 (2006): 456-469. Zheng, Yiqi, et al. "Relationships between photosynthesis and formaldehyde as a probe of isoprene emission." Atmospheric Chemistry and Physics 15.15 (2015): 8559-8576. Zheng, Yiqi, et al. "Drought impacts on photosynthesis, isoprene emission and atmospheric formaldehyde in a mid-latitude forest." Atmospheric Environment 167 (2017): 190-201.

---

## Author Response (AR1)

**Alves et al. Leaf phenology as one important driver of seasonal changes in isoprene emission in central Amazonia**

**1) Point-by-point response to the reviews**

**a. Reply to Referee 1**

The study by Alves et al. provides a context for phenological control over isoprene emissions in an Amazonian tropical forest. I think that the results are interesting, but only because this is a tropical forest. The potential for phenology and leaf age to control isoprene emission rates has been recognized in past studies going back 25 years. Studies by Fall, Monson, Harley, Litvak, Sharkey, Loreto and many others have clearly shown these relationships in temperate forest trees. The Alves et al. study is most interesting because it deals with a tropical forest, for which this type of insight is missing.

The main critique I level against the study is that it is written to largely ignore this past body of work, and the broader context of phenological influences over isoprene emission, and instead makes it sound like this is a new relationship discovered since 2013.

I recommend a major revision of the work that honestly takes into account the historical context of the phenology-emission relationship and its relevance to the observations made in the tropical forest. In this revision I recommend making it clear that the novel aspects of the current work are that it (1) is among the first to show the importance of the phenology-emission relationship in a tropical forest, and (2) that it allows for the MEGAN model to be modified to better predict emissions in tropical forests.

**Author's response**: We now understand that the novelty of this manuscript was not clear in the text. Indeed, leaf phenology as one important driver for seasonal changes in isoprene emission is only new here with respect to a tropical rainforest. We do not want nor intend to ignore the well-established literature focusing on temperate forests. Therefore, the Introduction has been rewritten citing the relevant temperate forest literature and putting it within the context of this study. We appreciate all the comments made by the referee, and we think the manuscript has been improved after considering and accepting his/her comments and suggestions.

**Referee's comment** - Lines 66-68. The phrase "as BVOC emissions are regarded as highly significant for ecosystem productivity (Kesselmeier et al., 2002) with isoprene being the most emitted hydrocarbon, it thereby plays an important role in carbon balance", is worded strangely. What does "highly significant for ecosystem productivity" mean? Why is that part of the phrase supported with a reference, but the next part of the phrase, "it thereby plays an important role in carbon balance", is not supported by a reference? Does the Kesselmeier reference cover both parts of the phrase? If not, it seems that a second reference is needed for the second part of the phrase. I am especially interested in what is meant by "important role", as my understanding is that isoprene emission occurs as a small absolute flux compared to overall carbon fluxes (e.g., approximately 1000 times lower).

**Author's response** – BVOC emissions are small when compared to net primary productivity and gross primary productivity, but the carbon emitted in form of BVOCs can be significant for net ecosystem productivity, and comparable to the magnitude of net biome productivity (Kesselmeier et al., 2002). Because isoprene is the most emitted BVOC, we suggest that its contribution to carbon balance is highly important when compared to other BVOCs. To make this clearer, these sentences have been rewritten in the manuscript.

**Author's changes in manuscript** – Line 64. "Moreover, isoprene emissions could play an important role in the carbon balance, because it is the most emitted within BVOCs, which are regarded as highly significant for net ecosystem productivity, with their losses comparable to the magnitude of net biome productivity (Kesselmeier et al., 2002); and carbon dioxide is believed to be the fate of almost half of the carbon released in the form of BVOCs (Goldstein and Galbally, 2007).

**Referee's comment** - Lines 402-406. The phrase "and as isoprene emissions are strongly dependent on leaf age and mainly emitted by mature leaves (Alves et al., 2014), seasonal changes in the forest leaf-age fractions may also influence the seasonality of isoprene emissions, suggesting higher emissions in the presence of more mature leaves and during high ecosystem photosynthetic capacity efficiency." I found this to be a bit of an egregious claim by the authors. The implication is that the dependence of isoprene emissions on leaf age and phenology was only discovered by the authors in 2014 ignores a rich literature that has shown the effects of leaf age and phenology on isoprene emissions going back at least two decades. There are many past studies showing, in explicit terms, the effects of leaf age and phenology on isoprene emissions. The authors seem gracious in citing the rich literature connecting photosynthate to isoprene emissions, but then take sole credit for discovering the connection between phenology and isoprene emissions. This should be corrected so that the true scope of the problem and related past research is brought honestly into this paper.

**Author's response** – This paragraph has been rewritten with more literature added.

**Author's changes in manuscript** – Line 402. "...and as isoprene emissions are strongly dependent on leaf ontogenetic stage  - due to the developmental patterns of isoprene synthase activity that gradually increases with leaf maturation and decreases with leaf senescence (Alves et al., 2014; Kuzma and Fall, 1993; Mayrhofer et al., 2005; Monson et al., 1994; Niinemets et al., 2004, 2010; Schnitzler et al., 1997) - seasonal changes in the forest leaf-age fractions may also influence the seasonality of isoprene emissions, suggesting higher emissions in the presence of more mature leaves and during high ecosystem photosynthetic capacity".

**Referee's comment** - Lines 428-432. The phrase "This is consistent with previous studies that provide evidence that alternative non-photosynthetic pathways may contribute to isoprene synthesis under stress (Loreto and Delfine, 2000), which may then lead to a decoupling of isoprene emission from photosynthesis at high temperatures (Foster et al., 2014)." This seems to be a rather large and speculative jump in logic. There is no reason to suspect that the seasonal offset between GPP and isoprene emission rate is due to the use of stored carbon sources. In fact, the past literature (going back into the 1990s) shows that the leaf age effect is likely due to developmental (ontogenetic) patterns of isoprene synthase activity. Thus, the phenological constraint over isoprene emission (1) has the potential to override the correlation between photosynthesis rate and isoprene emission rate, and (2) this is due to an enzymatic limitation, not a limitation of carbohydrate availability. The authors seem to be unaware of this past literature as it is not mentioned in their paper. This should be corrected.

**Author's response** – We agree with this comment, and we decided to remove this whole paragraph. The relation between leaf age and isoprene synthase activity is mentioned in another part of the manuscript. We understand that this fits in better at lines 401-410.

**Author's changes in manuscript** – Line 401. Photosynthesis supplies the carbon to the methyl erythritol phosphate pathway to produce isoprene (Delwiche and Sharkey, 1993; Harley et al., 1999; Lichtenthaler et al., 1997; Loreto and Sharkey, 1993; Rohmer, 2008; Schwender et al., 1997), and isoprene emissions are strongly dependent on leaf ontogenetic stage  - due to the developmental patterns of isoprene synthase activity that gradually increases with leaf maturation and decreases with leaf senescence (Alves et al., 2014; Kuzma and Fall, 1993; Mayrhofer et al., 2005; Monson et al., 1994; Niinemets et al., 2004, 2010; Schnitzler et al., 1997). Therefore, seasonal changes in the forest leaf-age fractions may also influence the seasonality of isoprene emissions, suggesting higher emissions in the presence of more mature leaves and during high ecosystem photosynthetic capacity efficiency.

**Referee's comment** - Lines 512-513. The phrase "However, less notable factors might also influence ecosystem isoprene emission." Once again, this phrase makes it seem like very few past studies have considered factors like phenology or leaf age as an important control over isoprene emissions. Actually, these factors have been recognized as just as important as temperature and light for over 25 years. The authors need to present their results in a way that embeds them honestly within the rich past tradition of isoprene emissions research.

**Author's response** – This sentence meant to say that less notable factors might also influence ecosystem isoprene emission in tropical forests. Leaf phenology, with notable seasonal changes in the Amazonian rainforest, was just recently discovered (Huete et al., 2006; Lopes et al., 2016; Myneni et al., 2007; Saleska et al., 2016; Wagner et al., 2017), and there is still some debate about it (e.g. Morton et al., 2014; Samanta et al., 2010). The fact that for many years seasonal changes and leaf phenology were not thought to be important for tropical forests, given their evergreen condition state, led the scientific modeling community to assume that leaf phenology affects very little forest and atmosphere gas exchanges in tropical forests. However, after remote sensing studies showed seasonal biomass changes (Myneni et al., 2007) and seasonal changes in isoprene emissions (Barkley et al., 2009, 2013), models were improved in order to better represent seasonal biomass changes and leaf age in tropical forests.  This is the case of MEGAN that uses variations in LAI to parameterize changes in leaf age, and then changes in the emission activity factor of isoprene emission (Guenther et al., 2012). However, because leaf phenology in tropical forests is not as notable as in temperate forests, some insights on how changes in leaf age over the year may affect seasonal isoprene emissions are still missing, and there is a lack of representation of this process in models. Here, we wanted to show that leaf phenology affects seasonal changes of isoprene emission and that is, in fact, new information for tropical forests.

[revised manuscript text omitted]

b. **Reply to Referee 2**

Alves et al. present a 7-month observation of isoprene flux in central Amazonia, and demonstrate the role of leaf age in controlling its seasonal variation. This study deserves documentation because it provides a long observational record of isoprene emissions and in-situ comonitored leaf phenology, which is scarce in the tropics. However, I agree with the other reviewer that this manuscript ignores a pool of previous literature. Some other issues may need to be addressed as well before accepted for publication.

**Author's response:** We thank all the comments and suggestions made by the referee.

In terms of the effect of leaf phenology on isoprene emission, we acknowledge that this is an important factor and that has been pointed out in past studies from temperate forests. Here, we wanted to show that this could also be important in tropical forests, which was not clearly shown before because seasonal changes in leaf age and leaf biomass in tropical forests are not as strong as in temperate forests. In addition, only recently has the leaf phenology in tropical forest, especially in the Amazon forests, been shown to be one important factor on forest physiological processes (Huete et al., 2006; Lopes et al., 2016; Myneni et al., 2007; Saleska et al., 2016; Wagner et al., 2017).

We understand that the main novelty of the results of this manuscript is due to our study region, a tropical forest, and we have tried to emphasize this point now. Moreover, previous literature on leaf phenology and isoprene emissions have also now been added.

**Referee's comment -** L64-68. What is the contribution of isoprene to total $CO_2$ emission in percentage? To my knowledge the number is very small.

**Author's response** – According to Guenther (2002), the percentage of carbon emitted as isoprene is about 1% to 4% at optimal temperatures for plant growth, but can exceed 10% at higher temperatures.

**Author's changes in manuscript** - Line 64. "Moreover, isoprene emissions could play an important role in the carbon balance, because it is the most emitted within BVOCs, which are regarded as highly significant for net ecosystem productivity, with their losses comparable to the magnitude of net biome productivity (Kesselmeier et al., 2002); and carbon dioxide is believed to be the fate of almost half of the carbon released in the form of BVOCs (Goldstein and Galbally, 2007).

**Referee's comment** - L81. "drivers of isoprene" should be "drivers of isoprene emissions".

**Author's response** – This sentence was rewritten as suggested by the referee.

**Author's changes in manuscript** – L79. "Some of these *in situ* studies indicate hat environmental factors such as solar radiation and temperature are primary drivers of isoprene emission..."

**Referee's comment** - L83-88. "canopy phenology could therefore be an important seasonal driver". Does the phenological control on isoprene emissions only occur through photosynthesis? Kuzma and Fall, 1993 suggested that the enzyme activity regulates the isoprene emission in response to leaf development. The authors may want to replace the sentence with a paragraph of literature review (including mid-latitude studies) on the theory and observations of isoprene emission versus leaf phenology. See review paper Harrison et al., 2013 (Table S2) and many others, Niinemets, Monson, Sharkey, etc.

**Author's response** – We agree, and we have rewritten the paragraph giving information on how isoprene emission can be affected by leaf age and ontogeny.

**Author's changes in manuscript** – L85. "However, besides long-term seasonal variation in light and temperature, other biological factors might act on seasonal changes of isoprene emission, as the case of canopy phenology. Previous studies with temperate species have shown that isoprene emission capacity is affected by leaf age and ontogeny (Kuzma and Fall, 1993; Mayrhofer et al., 2005; Monson et al., 1994), because: (1) isoprene synthase and other enzymes of isoprene synthesis pathway (MEP pathway)

depends on the leaf ontogeny - isoprene synthase activity is low or absent in very young leaves, increasing gradually until full leaf maturation, and decreasing with leaf senescence (Schnitzler et al., 1997); (2) for species of non-senescent leaves, or with a life-span of more than one year, foliage shading and time-dependent changes of physiological activity of leaves could decrease isoprene emission capacity (Niinemets et al., 2004, 2010); and (3) leaf structure varies with leaf ontogenetic stage, indicating that seasonal isoprene emission capacity is affected by seasonal structural changes in leaves (Niinemets et al., 2004, 2010)".

**Referee's comment - L132**: How did you choose the 5 or 6 days every month for measurement? What are the cloud conditions?

**Author's response** – Days were chosen between $20^{th}$ and $30^{th}$ of each month. When possible, measurements were carried out on very sunny days and without rain. But, a few days in June and October were a little cloudy. Cloud conditions can change very quickly in the Amazon. Therefore, to really characterize differences in isoprene emissions between sunny and cloudy days, more long-term measurements are needed.

**Referee's comment** - In 2013, the monthly variation of satellite-derived isoprene emission is totally wrong compared to in-situ measurement. Is it because you only have a few days' measurement each month? I suggest to include the 2013 satellite isoprene curve in Figure 3 for a direct comparison. Include both monthly average and the REA period average.

**Author's response** – We think part of the differences between satellite-derived isoprene emission and *in situ* isoprene emission is due to the smaller number of days of *in situ* measurements. But, differences due to the spatial resolution should also be considered. Satellite-derived isoprene emission resolution is 50 km, whereas *in situ* measurements have a much smaller footprint. This might suggest that *in situ* measurements have more impact from local effects, which could be diminished when lower spatial resolution is being analyzed.

**Author's changes in manuscript** - Satellite-derived isoprene flux was added to Figure 3.

[Figure]

Figure 3: (a) Monthly cumulative precipitation given by the Tropical Rainfall Measuring Mission (TRMM) for the K34 tower domain in 2013. (b) Monthly averages of PAR and (c) air temperature, both measured every 30 minutes during 6:00-18:00h, local time, at the K34 tower site in 2013. (d) Isoprene flux measured with the REA system at the K34 tower site in 2013; and OMI satellite-derived isoprene flux for the K34 tower region.

**Text in the manuscript:** L378. "The reasons why satellite-derived isoprene fluxes are weakly correlated to ground-based isoprene fluxes can be attributed to either the difference in the studied scales (e.g. local effects could have major influences on ground-based isoprene fluxes) and/or the uncertainties associated with the methodologies used to estimate or calculate fluxes. The high correlation between satellite-based fluxes and air temperature or PAR is not unexpected, because higher temperatures and solar radiation fluxes favor isoprene emissions. Note however that the satellite-derived fluxes might also be subject to inherent uncertainties, due to the existence of other HCHO sources, in particular biomass burning (during the dry season) and methane oxidation. Since these latter contributions are favored by high temperature and radiation levels, they could possibly contribute to the high correlation found between satellite-based isoprene and meteorological variables".

**Referee's comment** - L333: Does Table 2 show R or $R^2$? "Explaining 59% of variations" usually refers to $R^2$ values. The abstract should be consistent, too.

**Author's response** – Table 2 shows $R^2$ values. The corresponding sentences in the abstract are written with $R^2$ values inside brackets, for example in L44 "…the highest correlation with observed isoprene flux seasonality ($R^2$=0.59, $p$<0.05)", and L50 "…significantly improved simulations in terms of seasonal variations of isoprene fluxes ($R^2$=0.52, $p$<0.05)".

**Referee's comment** - L342: "Regression" should be "Correlation".

**Author's response** – In this case, it is really regression, because this is what is shown in Table 2.

**Referee's comment** - L352-362, Figure 6: No matter with EAF changes or not, the MEGAN monthly variations look more similar to the satellite-derived isoprene emissions. Again, is this because the in-situ observation only includes a few days every month? I wonder whether these days can represent emissions during the whole month. Is MEGAN run at a day- by-day basis? If so, the authors may try take out the MEGAN simulations during the same days as the REA measurement to see whether the correlations are improved.

**Author's response** – The reason why MEGAN estimates and *in situ* observations have low correlation is, probably, in part due to the small number of *in situ* observations. However, when comparing results of MEGAN estimates with the same days of *in situ* observations, we did not improve the correlations. One issue is that, for a few days in July and December, there were gaps in the PAR and temperature datasets, which prevented us from simulating isoprene flux for those days. Therefore, a correlation between MEGAN estimates and *in situ* observations for the same days of REA measurements is not possible.

For verification, the bellow figure shows an inset panel with MEGAN estimates of the same days of *in situ* observations:

[Figure]

Figure 5: Isoprene flux observed (REA) and estimated with MEGAN 2.1 in default mode, leaf age algorithm driven by MODIS-LAI, and with MEGAN 2.1 leaf age algorithm driven by CAMERA-LAI. EAF stands for emission activity factor, which was changed for the different leaf age classes based on emissions of *E. coriacea* (Alves et al., 2014). The inset panel shows the four MEGAN simulations only for the days of REA measurements.

Another possibility is soil-moisture dependence. Quite a few studies showed the importance of water availability, e.g. Pegoraro et al., 2004-2006, Zheng et al., 2015, 2017, etc. In Figure 3, observed isoprene flux shows a similar monthly pattern as the TRMM precipitation in dry and dry-to-wet seasons (when water is limited). The authors may want to do a MEGAN sensitivity test that includes soil moisture dependence or at least discuss the role of soil moisture in Section 4.1.

**Author's response** – We do not have soil moisture data simultaneous to our REA measurements. For this experimental site, a previous study showed that during the dry season there is only a small reduction (~10 %) in soil moisture compared to the wet season (Cuartas et al., 2012); this reduction does not induce water stress to this forest region (Wagner et al., 2017). Moreover, based on the dataset of soil moisture shown from 2002 to 2006 (Cuartas et al., 2012), the soil moisture always exceeds the threshold for the isoprene drought response in MEGAN 2.1 (Guenther et al., 2012), which means that MEGAN would predict that there are no variations in isoprene emissions due to these observed changes in soil moisture. Therefore, we feel justified in having kept the soil moisture constant in model simulations.

**Referee's comment** - L434: The wording "during leaf phenology" is strange.

**Author's response** – This sentence was rewritten to "…isoprene emission during leaf ageing". However, after doing some revisions based on the comments from the other referee, we removed this paragraph and wrote a new one with more relevant information from previous studies of temperate forests.

**Author's changes in manuscript** – "However, besides long-term seasonal variation in light and temperature, other biological factors might act on seasonal changes of isoprene emission, as the case of canopy phenology. Previous studies with temperate species have shown that isoprene emission capacity is affected by leaf age and ontogeny (Kuzma and Fall, 1993; Mayrhofer et al., 2005; Monson et al., 1994), because: (1) isoprene synthase and other enzymes of isoprene synthesis pathway (MEP pathway) depends on the leaf ontogeny - isoprene synthase activity is low or absent in very young leaves, it increases gradually until full leaf maturation, and decreases with leaf senescence (Schnitzler et al., 1997); (2) for species of non-senescent leaves, or with life-span of more than one year, foliage shading and time-dependent changes of physiological activity of leaves could decrease isoprene emission capacity (Niinemets et al., 2004, 2010); (3) and leaf structure varies with leaf ontogenetic stage, indicating that seasonal isoprene emission capacity is affected by seasonal structural changes in leaves (Niinemets et al., 2004, 2010)".

**Referee's comment** - Figure 2, 3, 6: As a convention, the panel numbers (a)(b)(c) are usually placed in front of description.

**Author's response** – Suggestion accepted. The panel numbers are now placed in front of the description for each of these figures.

**Author's changes in manuscript** -

**Figure 2**. (a) Monthly averages of photosynthetic active radiation (PAR) and (b) air temperature from 2005 to 2013 at the K34 tower site (measured every 30 min during - 6:00-18:00h, local time). (c) OMI satellite-derived isoprene flux in a resolution of 0.5º centered on K34 tower site from 2005 to 2013. Monthly averages of isoprene flux were scaled to 10:00-14:00, local time. Error bars represent one standard error of the mean.

**Figure 3**. (a) Monthly cumulative precipitation given by the Tropical Rainfall Measuring Mission (TRMM) for the K34 tower domain in 2013. (b) Monthly averages of PAR and (c) air temperature, both measured every 30 minutes during 6:00-18:00h, local time, at the K34 tower site in 2013. (d) Isoprene flux measured with the REA system at the K34 tower site in 2013.

**Figure 6**. (a) Emission activity factor (EAF) of isoprene for each leaf age class assigned in the default mode of MEGAN 2.1 proportional to leaf age class distribution derived from field observations (CAMERA-LAI). (b) Isoprene EAF for each leaf age class, obtained from leaf level measurements of the tree species *E. coriacea,* proportional to leaf age class distribution derived from field observations (CAMERA-LAI). Observations of the tree species *E. coriacea* (Alves *et al.*, 2014) and CAMERA-LAI are both from the K34 site.

**Referee's comment** - Some references: Kuzma, Jennifer, and Ray Fall. "Leaf isoprene emission rate is dependent on leaf development and the level of isoprene synthase." Plant physiol- ogy 101.2 (1993): 435-440. Harrison, Sandy P., et al. "Volatile isoprenoid emissions from plastid to planet." New Phytologist 197.1 (2013): 49-57. Niinemets, Ülo, et al. "The emission factor of volatile isoprenoids: stress, acclimation, and developmental responses." Biogeosciences 7.7 (2010): 2203. Pegoraro, E., et al. "Effect of drought on isoprene emission rates from leaves of Quercus virginiana Mill." Atmospheric Envi- ronment 38.36 (2004): 6149-6156. Pegoraro, Emiliano, et al. "The interacting effects of elevated atmospheric CO 2 concentration, drought and leaf-to-air vapour pressure deficit on ecosystem isoprene fluxes." Oecologia 146.1 (2005): 120-129. Pegoraro, Emiliano, et al. "Drought effect on isoprene production and consumption in Biosphere 2 tropical rainforest." Global Change Biology 12.3 (2006): 456-469. Zheng, Yiqi, et al. "Relationships between photosynthesis and formaldehyde as a probe of isoprene emission." Atmospheric Chemistry and Physics 15.15 (2015): 8559-8576. Zheng, Yiqi, et al. "Drought impacts on photosynthesis, isoprene emission and atmospheric formaldehyde in a mid-latitude forest." Atmospheric Environment 167 (2017): 190-201.

**Author's response**: We thank the suggestion of the above references, and we have added some to the manuscript.

**References**

[revised manuscript text omitted]

K34 site.

Eliane Alves 6/1/18 2:33 PM